# "Satanism is witchcraft's younger sibling": Changing perceptions of natural and supernatural anaemia causality in Malawian children

**Sarah Svege**[1]*, **Thandile Nkosi-Gondwe**[1,2], **Siri Lange**[3,4]

1 Centre for International Health, Department of Global Public Health and Primary Care, University of Bergen, Bergen, Norway, 2 School of Public Health & Family Medicine, College of Medicine, University of Malawi, Blantyre, Malawi, 3 Department of Health Promotion and Development, University of Bergen, Bergen, Norway, 4 Chr. Michelsen Institute, Bergen, Norway

* sarah.svege@icloud.com

**Data Availability Statement:** All relevant data are within the paper and its Supporting Information files.

## Abstract

In countries of sub-Saharan Africa, many children are admitted to hospital with severe forms of anaemia. The late hospital admissions of anaemic children contribute significantly to child morbidity and mortality in these countries. This qualitative study explores local health beliefs and traditional treatment practices that may hinder timely seeking of hospital care for anaemic children. In January of 2019, nine focus group discussions were conducted with 90 participants in rural communities of Malawi. The participants represented four groups of caregivers; mothers, fathers, grandmothers and grandfathers of children under the age of five. The Malawian medical landscape is comprised of formal and informal therapeutic alternatives–and this myriad of modalities is likely to complicate the healthcare choices of caregivers. When dealing with child illness, many participants reported how they would follow a step-by-step, 'multi-try' therapeutic pathway where a combination of biomedical and traditional treatment options were sought at varying time points depending on the perceived cause and severity of symptoms. The participants linked anaemia to naturalistic (malaria, poor nutrition and the local illnesses *kakozi* and *kapamba*), societal (the local illness *msempho*) and supernatural or personalistic (witchcraft and Satanism) causes. Most participants agreed that anaemia due to malaria and poor nutrition should be treated at hospital. As for local illnesses, many grandparents suggested herbal treatment offered by traditional healers, while the majority of parents would opt for hospital care. However, participants across all age groups claimed that anaemia caused by witchcraft and Satanism could only be dealt with by traditional healers or prayer, respectively. The multiple theories of anaemia causality combined with extensive use of and trust in traditional and complementary medicine may explain the frequent delay in admittance of anaemic children to hospital.

**Funding:** The study was funded by the Research Council of Norway through the Global Health and Vaccination Programme (GLOBVAC), project number 234487. GLOBVAC is part of the EDCTP2 programme supported by the European Union. The Council had no role in study design, data collection and analysis, decision to publish, or preparation of the manuscript.

**Competing interests:** The authors have declared that no competing interests exist.

## 1. Introduction

Low levels of haemoglobin in the blood is known as anaemia, and contributes significantly to morbidity and mortality among young children in sub-Saharan Africa (SSA). The most recent WHO estimates report that more than half of preschool-age children in African regions have some form of anaemia, with 3.6% affected by severe anaemia [1]. The major causes of anaemia in these areas are iron deficiency, malaria, hookworm and hereditary blood disorders [2]. In SSA, many children are admitted to health facilities with severe anaemia that may have been avoided if medical attention was sought earlier [3]. In order to design and implement large-scale and long-standing health interventions one should be familiar with the local concepts of health, illness and therapeutic traditions at community level, as well as the factors that influence care-seeking behaviour.

There has been increasing interest from scholars in public health to understand care-seeking for child illness in low-income countries. A qualitative review on household recognition and response to child illness in SSA by Colvin et al. describes care-seeking as a process consisting of four different modes: 1) direct caregiver recognition and response, 2) seeking advice and negotiating access in the family, 3) making use of the "middle layer" of community-based treatment options, and 4) accessing formal medical services at a clinic or hospital [4]. Colvin et al. suggest that the "middle layer" includes actors such as caregivers, traditional healers and community health workers.

For many millions of people, herbal medicines, traditional treatments and traditional practitioners are the main source of healthcare–and sometimes the only accessible, affordable and culturally acceptable source of care [5]. The WHO Traditional Medicine Strategy aims to promote universal health coverage by supporting member states in improving and, if appropriate, integrating traditional and complementary medicine (T&CM) within national healthcare systems [5]. In many African countries, the extensive use of T&CM can be explained by i) the lack of available formal healthcare, ii) services costing less than those provided by formal healthcare, iii) trust towards healers or 'spiritual' doctors within the community and iv) its perceived superiority and efficacy in handling certain conditions [6, 7]. T&CM, often offered by informal healthcare providers, is common globally and is typically distinctly separated from 'modern' biomedicine. It has however been recognised that biomedicine and what is often termed 'traditional' medicine do not constitute binary opposites, but are both part of a continuum, and that each of them are perceived to be useful for different kinds of disorders [8]. Herbs used in traditional medicine are incorporated in biomedicine, and non-medical therapies such as acupuncture have been practised by some medical doctors [9]. At the same time, Western medical science do put a fundamental opposition between the 'real and unreal' [10], and in most countries there are hierarchies of legitimacy in the way different practitioners are perceived [11, 12]. Services provided by health practitioners without formal training are generally not accepted as valid by health ministries, scientists and physicians. Still, the use of informal or 'traditional' health practitioners that 'operate on the margins of legitimacy' is widespread and should thus not be neglected [12]. Medical pluralism [9] or medical parallelism [13] is evident in many African communities, and may expand the possible avenues of treatment, but also complicate the decision-making processes for patients and their families. Colvin et al. conclude that "treatment decision making is a dynamic process characterised by uncertainty and debate, experimentation with multiple and simultaneous treatments, and shifting interpretations of the course of the illness and best treatment options" [4]. The review also highlights research gaps such as lack of evidence on the role of religious or supernatural forces in health beliefs and practices [4]. Previous studies have explored local concepts of anaemia among caregivers in Côte d'Ivoire, Tanzania and Uganda [14–16]. In these studies,

nutritional deficiency and malaria were frequently reported causes of anaemia. Supernatural forces such as witchcraft and evil spirits were also mentioned, but not thoroughly described. This study contributes to filling this research gap by focusing specifically on the perceived supernatural causes of anaemia. Additionally, it investigates other perceived causes of anaemia and the available treatment options in a Malawian context. We will use a framework provided by Foster and Kahissay et al. to distinguish between naturalistic, societal and supernatural or personalistic aetiologies of anaemia [17, 18]. To our knowledge, this is the first study looking at caregiver perceptions on anaemia causality in Malawi.

## 2. Methods

### 2.1 Study setting

This qualitative study is nested in a clinical trial on delivery of malaria chemoprevention as post-discharge management for children with severe anaemia in Malawi (NCT02721420). The main trial and its sub-studies are performed in collaboration with the College of Medicine at University of Malawi [19]. Malawi is a country in Southeastern Africa with approximately 19 million inhabitants. It is ranked as 174 of 189 countries on the Human Development Index and has the world's second lowest GDP per capita [20]. The under-five mortality rate is 63 deaths per 1,000 live births, which can be translated to 1 of every 16 children dying before their fifth birthday [21]. Under-five mortality is higher in rural areas than urban areas with 77 versus 61 deaths per 1,000 live births [21]. Overall, 63% of children aged 6–59 months suffer from some degree of anaemia; 27% were classified with mild anaemia, 34% with moderate anaemia and 2% with severe anaemia [21]. Anaemia prevalence is higher among children in rural areas (64%) compared to urban areas (56%), and it declines with increasing household wealth and mother's education [21]. The Malawi health service delivery system is pyramidal, consisting of tertiary, secondary, primary and community care levels. District and central hospitals provide secondary and tertiary care services, respectively. Primary care is delivered through clinics and health centers where curative, maternity and preventive services are offered. At community level, childhood illnesses are managed through the integrated community case management (iCCM) approach [22]. The fieldwork for this study was conducted in the Zomba district of Southern Malawi where the main trial was also performed. In this district, the majority of men and women aged 15–49 years work within the agricultural sector [21]. A recent study identified geographical distance and travel expenses as important barriers to healthcare-seeking in Zomba [23].

### 2.2 Data collection

Nine focus group discussions (FGDs) were conducted with caregivers in Malawi in January 2019. The aim of the FGDs was to learn about the experiences and perceptions of participants, and give them an opportunity to share and compare their thoughts. The FGDs also allowed observation of group dynamics and interaction between participants. The number of FGDs was pre-defined based on budgetary restrictions, but briefing sessions within the research team showed that saturation was achieved.

The study participants were recruited by local health workers. During recruitment, they were asked to take part in a discussion about childhood illnesses in their community. The FGDs were held at schools or local health facilities in rural and semi-rural villages. These were familiar meeting points in close proximity to the homes of participants. The discussions were held in the local language Chichewa by a Malawian moderator with extensive experience from fieldwork in similar study settings. Each discussion lasted approximately 1½-2 hours. Refreshments were served and each participant was given a fixed amount of 2000 MWK (2.5 USD) as

compensation for their time. The first author was present during all FGDs and briefing sessions.

A guide of open-ended questions was developed by the first and last author to explore topics of interest. The guide was informed by preliminary knowledge on local perceptions of anaemia from a previous study where we interviewed caregivers of children who had been hospitalised for severe anaemia [23]. Topic one and two in the guide were local illnesses that caregivers had associated with anaemia; namely 'kapamba' and 'kakozi'. Topic three, 'Satanism', and topic four, 'witchcraft', were supernatural forces that caregivers had reported as possible causes of anaemia. Topic five was 'the role of traditional healers', topic six was the genetic blood disorder 'sickle cell anaemia', and topic seven was 'treatment and prevention of anaemia'. To facilitate openness and reduce the social desirability effect, each topic was introduced in a neutral and non-judgemental way. For example, when asking about local illnesses, the moderator said: "two years ago we were in this area and some parents told us about the problem of kapamba, but we did not fully understand. In this group, can you please describe what kapamba is?" When asking about Satanism, the moderator said: "another serious problem that people told us about is Satanism. How would you describe Satanism?" The guide was pretested, reviewed and revised to meet the research objectives and time frame. A questionnaire survey on demographic characteristics was also completed by all participants.

## 2.3 Sample selection

The study participants (n = 90) were purposely selected according to gender, age, social role (grandparent/parent) and area of residence (rural/semi-rural). In each group, 9–11 caregivers took part in the discussion. Criteria for inclusion was to be a parent or grandparent of a child below five years of age. Specifically, caregivers were recruited to four different groups; mothers (three FGDs), fathers (two FGDs), grandmothers (two FGDs) and grandfathers (two FGDs). None of the participants were from the same household. In Malawi, mothers are often the primary caregivers and key decision-makers regarding their child's health. But they may also seek advice or permission from their husbands and parents/in-laws regarding healthcare decisions. Previous studies have shown how grandparents, and especially grandmothers, play an important role in healthcare decision-making at household level in Malawi [24] and other countries like South Africa, Mali and Tanzania [7, 25, 26]. The FGDs were composed based on gender and age to examine possible differences in gender and generational perspectives, and because FGDs normally function better with relatively homogenous groups [27]. A complete FGD overview can be seen in Table 1.

**Table 1. FGD overview.**

| FGD number (n = 9) | Type of village | Informant group | Number of participants (n = 90) |
|---|---|---|---|
| FGD 1 | Semi-rural | Mothers | 9 |
| FGD 2 | Rural | Fathers | 10 |
| FGD 3 | Rural | Grandmothers | 10 |
| FGD 4 | Semi-rural | Grandmothers | 10 |
| FGD 5 | Semi-rural | Fathers | 10 |
| FGD 6 | Semi-rural | Mothers | 11 |
| FGD 7 | Semi-rural | Grandfathers | 10 |
| FGD 8 | Rural | Mothers | 10 |
| FGD 9 | Rural | Grandfathers | 10 |

### 2.4 Data analysis

Audio file recordings of the FGDs were transcribed word-by-word in Chichewa by a research assistant who also translated them to English. Selected transcripts were thoroughly checked by the second author to ensure coherence between i) tape recordings and Chichewa transcripts, and ii) Chichewa transcripts and English translations. Data material from the FGDs was structured, coded and analysed in the data analysis software programme NVivo Version 12. The coding system consisted of main codes derived from the topic guide, as well as sub-codes that emerged during transcript readings. Text segments were attached to appropriate codes in order to categorise, compare and contrast the data. Next, themes and sub-themes were developed to further interpret and organise the findings. When we report our results, the focus will be on issues related to child health and anaemia even though the participants also discussed how witchcraft and Satanism may affect adults.

### 2.5 Ethical considerations

Ethical approval was granted from the College of Medicine Research and Ethics Committee (COMREC) in Malawi (P.02/15/1679) and the Regional Committees for Medical and Health Research Ethics in Norway (2015/537/REK vest). Informed written consent was obtained from all participants. Prior to signing the consent form with written letters or a thumbprint, the entire consent form was read to or by the participant. All participants were assigned a personal study ID to ensure confidentiality. The data is securely stored with access limited to members of the study team.

## 3. Results

This section will first briefly describe participant characteristics. Next, the participants' general perceptions of anaemia symptoms and treatment strategies will be presented. The main part of this section focuses on the perceived naturalistic, societal and personalistic causes and conditions related to anaemia in children.

### 3.1 Demographic characteristics of study participants

All study participants completed a short questionnaire survey prior to the FGD. The age range of the participants was 18 to 89 years. The survey showed that 79% worked in the agricultural sector as subsistence farmers with or without commercial crop-sale. Less than 4% had electricity in their homes, and 84.4% had earth or sand as flooring material. Close to all participants (96.7%) were Christians, and 87.8% stated that they were married. A little more than a third of the participants (34.4%) reported grade 7–9 as educational attainment, while 14.4%–all of them grandparents–never received any formal schooling. Only 12.2% of the study population had attended secondary or tertiary school above grade 10, and this group consisted mainly of young fathers. Tables 2 and 3 provide a detailed overview of demographic and household characteristics.

### 3.2 General perceptions of anaemia symptoms and treatment strategies

In Chichewa, low blood levels (anaemia) is referred to as 'lack of blood in the body' (*kuchepa kwa magazi nthupi*). Children with low blood levels were described by study participants as 'fair in complexion' with pale or yellow feet, hands and eyes:

> *"A child with anaemia looks very sick; the eyes are white, the body is yellow and the down side of the feet turns yellow." (Mother, FGD 8)*

**Table 2. Demographic characteristics of informants.**

| Study population | Mothers (n = 30) | Grandmothers (n = 20) | Fathers (n = 20) | Grandfathers (n = 20) | Total (n = 90) |
|---|---|---|---|---|---|
| | n (%) | n (%) | n (%) | n (%) | n (%) |
| **Civil status** | | | | | |
| Married | 27 (90) | 12 (60) | 20 (100) | 20 (100) | 79 (87.8) |
| Divorced/separated | - | 3 (15) | - | - | 3 (3.3) |
| Widow/widower | 2 (6.67) | 5 (25) | - | - | 7 (7.8) |
| Single | 1 (3.33) | - | - | - | 1 (1.1) |
| **Occupation** | | | | | |
| Farming | 17 (56.7) | 20 (100) | 15 (75) | 19 (95) | 71 (79) |
| Business & sales | 8 (26.7) | - | 3 (15) | - | 11 (12.2) |
| Labour-work | - | - | 2 (10) | 1 (5) | 3 (3.3) |
| Home-keeper | 5 (16.6) | - | - | - | 5 (5.5) |
| **Educational attainment (grade)** | | | | | |
| 0 | - | 5 (25) | - | 8 (40) | 13 (14.4) |
| 1–3 | - | 7 (35) | - | 7 (35) | 14 (15.6) |
| 4–6 | 12 (40) | 3 (15) | 2 (10) | 4 (20) | 21 (23.3) |
| 7–9 | 16 (53.3) | 5 (25) | 9 (45) | 1 (5) | 31 (34.4) |
| $\geq$10 | 2 (6.7) | - | 9 (45) | - | 11 (12.2) |
| **Ethnicity** | | | | | |
| Chewa | 21 (70) | 8 (40) | 4 (20) | - | 33 (36.7) |
| Lomwe | 2 (6.7) | 10 (50) | 11 (55) | 20 (100) | 43 (47.8) |
| Yao | 1 (3.3) | 2 (10) | 1 (5) | - | 4 (4.4) |
| Ngoni | 2 (6.7) | - | - | - | 2 (2.2) |
| Nyanja | 4 (13.3) | - | 4 (20) | - | 8 (8.9) |
| **Religious affiliation** | | | | | |
| Christianity | 30 (100) | 20 (100) | 17 (85) | 20 (100) | 87 (96.7) |
| Islam | - | - | 3 (15) | - | 3 (3.3) |

Swelling, or oedema, was another reported sign of anaemia:

*"They become fair or sometimes they may swell up and you know that the child is not well, that is anaemia."* (Mother, FGD 1)

Other reported symptoms were dizziness, fainting and lack of energy, and a few participants explained that low blood levels may lead to body aches:

*"They change, they get lighter in complexion and they also have body aches."* (Mother, FGD 6)

Some participants highlighted the importance of accessing hospital care *before* home-based or traditional treatment when a child is suffering from anaemia:

*"When a child or an adult gets malaria or anaemia, we need to run to the hospital first. If the hospital staff fails, that is when we go to seek traditional healers."* (Mother, FGD 1)

Other participants described how they, in cases of what they perceived as mild anaemia, would initially try to restore the blood levels at home by giving their children vegetables, watery fruits or boiled water with avocado pear leaves. Another home-based treatment is

**Table 3. Household characteristics of informants.**

| Characteristics | Study population in rural areas (n = 40) n (%) | Study population in semi-rural areas (n = 50) n (%) | Total (n = 90) n (%) |
|---|---|---|---|
| **Number of people in the household** | | | |
| 1–3 | 9 (22.5) | 21 (42) | 30 (33.3) |
| 4–6 | 29 (72.5) | 23 (46) | 52 (57.8) |
| ≥7 | 2 (5) | 6 (12) | 8 (8.9) |
| **Electricity** | | | |
| Yes | 0 | 3 (6) | 3 (3.3) |
| No | 40 (100) | 47 (94) | 87 (96.7) |
| **Rooms used for sleeping** | | | |
| 1 | 10 (25) | 6 (12) | 16 (17.8) |
| 2 | 17 (42.5) | 20 (40) | 37 (41.1) |
| ≥3 | 13 (32.5) | 24 (48) | 37 (41.1) |
| **Cooking fuel** | | | |
| Wood | 40 (100) | 43 (86) | 83 (92.2) |
| Charcoal | 0 | 7 (14) | 7 (7.8) |
| **Cooking location** | | | |
| Inside | 1 (2.5) | 1 (2) | 2 (2.2) |
| Outdoors | 2 (5) | 15 (30) | 17 (18.9) |
| Separate building | 37 (92.5) | 34 (68) | 71 (78.9) |
| **Flooring material** | | | |
| Earth/sand | 37 (92.5) | 39 (78) | 76 (84.4) |
| Cement | 3 (7.5) | 11 (22) | 14 (15.6) |
| **Time to obtain water** | | | |
| Water on premises | 12 (30) | 14 (28) | 26 (28.9) |
| <30 min | 19 (47.5) | 24 (48) | 43 (47.8) |
| >30 min | 9 (22.5) | 12 (24) | 21 (23.3) |
| **Phone ownership** | | | |
| Yes | 24 (60) | 36 (72) | 60 (66.7) |
| No | 16 (40) | 14 (28) | 30 (33.3) |

boiled water mixed with leaves from various trees (*vinyo*, *chidede*, *mpinjipinji*, *nyowe*) that produce a red colour which resembles blood. If available, sugar is added to reduce the bitter taste before the mixture is given to the child. Some caregivers explained how they would prepare the child's porridge or tea with powder from crushed tree leaves. Treatment-seeking behaviour and decision-making processes were related to reported level of formal education. Participants with secondary education (grades 9–12), of whom were mostly fathers, would emphasise the importance of accessing hospital care as a primary strategy. They spoke avidly about the 'professionalism' and 'skills' of medical doctors and health personnel, and many of them would only resort to traditional medicine if hospital medicine failed. On the contrary, those with limited or no formal education, of whom were mainly grandfathers and grandmothers, strongly advocated the use of home-based herbal remedies. They also explained how they would initially seek help and guidance from traditional healers and community elders prior to accessing hospital as a secondary strategy. However, for more serious cases of anaemia, participants across multiple FGDs agreed that hospital treatment, such as a blood transfusion, is the most efficient treatment option. Still, as will become evident below, the biomedical treatment is only perceived to be effective in cases of anaemia due to 'natural' causes such as malaria or nutritional deficiency.

### 3.3 Naturalistic disease aetiologies

When asked about the possible causes of anaemia in children, participants frequently reported malaria infection and poor nutrition. In medical science, these are considered scientifically acknowledged, natural or biomedical causes of anaemia. Two local illnesses known as *kakozi* and *kapamba* were also reported as possible anaemia aetiologies.

**Malaria.** Most participants acknowledged the association between malaria and anaemia:

*"When a child has malaria, it may happen that the malaria parasites are so many that they attack the blood, and since blood is where the strength comes from, it leads to anaemia. That is why they advise us to sleep under bed-nets." (Mother, FGD 1)*

In addition to sleeping under bed-nets, the participants talked about how malaria-related anaemia can be prevented by not playing or walking in stagnant, dirty water which is a typical breeding site for Anopheles mosquitoes.

**Poor nutrition.** Nutritional deficiency, or lack of food groups in the child's diet, was repeatedly brought up as a cause of low blood levels. Several caregivers emphasised the importance of incorporating all 'six food groups' in the child's diet:

*"A child who has reached the eating stage is supposed to eat all the six food groups, like nsima, beans, meat, fruit such as bananas, fish, vegetables and even porridge mixed with groundnuts and many other things." (Mother, FGD 1)*

Although knowledge about the 'six food groups' was more prevalent among mothers, a few grandparents also mentioned it:

*"Not eating a balanced diet can cause anaemia, for example if a child does not have all the six groups of food they may lack some blood." (Grandmother, FGD 3)*

Additionally, some participants stated that nutritional deficiency and anaemia could be the result of a child's lack of appetite due to *kakozi* and *kapamba*. In both of these local illnesses, anaemia symptoms were not necessarily the main symptoms, but rather one among several symptoms.

**Kakozi.** Caregivers described *kakozi* as a shrinking of genitals seen mostly in young boys whose testicles leave the scrotum and ascend upward. The illness was linked to cold weather, and no participants claimed that it could be caused by evil forces. The child usually experiences stomach aches, an urgency to urinate and a burning, piercing pain while urinating. Based on these symptoms *kakozi* may resemble a urinary tract infection. Additionally, it was reported that *kakozi* can give rise to symptoms that are seen in malaria patients such as fever and chills:

*"Kakozi usually attacks the boys. They shiver like they have malaria, the face turns light and when you check them you find that their testicles have gone inside so they have stomach pains and cry. If the illness strikes hard, they also pass out blood when urinating." (Grandmother, FGD 4)*

Some caregivers mentioned frequent urination as a common symptom of *kakozi*, and they emphasised how loss of fluid during urination may lead to anaemia:

*"From what we know, most of the blood is water so when you urinate often you will lose blood and become anaemic." (Father, FGD 5)*

*Kakozi* was also related to loss of appetite and a subsequent development of anaemia:

*"A child with kakozi gets fever often and does not eat–this makes the child not produce enough blood, and when you go to the hospital they find the child to be anaemic. But even when they treat anaemia, the child still gets anaemia again. That is when the elders say; this is kakozi."* (Mother, FGD 1)

According to participants, children are sometimes given malaria treatment at the hospital although community members believe they are actually suffering from *kakozi*:

*"At the hospital they say it is malaria since the child has fever and they treat with malaria drugs, but that does not deal with the actual illness because it comes back later."* (Grandmother, FGD 4)

Participants across multiple FGDs stated that hospital treatment of *kakozi* only provides a short-lived absence of pain, but no complete healing. For this reason, they advised the use of traditional remedies from healers:

*"Firstly, we go to the traditional healer because when your child is sick you show your friends first. If they know and tell you that it is kakozi we run for local help and not Western help since the Western help is for malaria."* (Grandmother, FGD 4)

One traditional treatment practice for *kakozi* is the mixing of boiled water with various herbs and plants to make a medicine known as *chikanje*. The medicine is either served as a liquid or inserted through small cuts in the child's skin.

**Kapamba.** *Kapamba* is the Chichewa term for spleen, but the participants were divided in their perception of what *kapamba* is and how it can make people ill. One participant, a grandmother, described it as the organ responsible for blood filtering, while some viewed *kapamba* as the kidney or an illness that affects the kidney. A young father characterised *kapamba* as an accumulation of clotted blood that may prevent the passage of blood to other areas of the body:

*"Kapamba is blood that clots and forms a tumour inside, so since it is a process where the blood is coming together in one place, that causes anaemia."* (Father, FGD 5)

Most of the participants defined *kapamba* as a 'tumour' that drains blood and causes pain:

*"Kapamba is indeed a tumour and because of that tumour, the blood does not flow on one side, and that is why the child has yellow skin which is a sign that it sucks blood."* (Mother, FGD 8)

Specifically, children with the 'tumour version' of *kapamba* will bend towards the affected side and struggle with standing up straight due to sharp pain. As the tumour drains blood from other parts of the body, it reportedly causes anaemia symptoms such as pale skin and fatigue:

*"A child with kapamba looks lighter in complexion, looks gloomy and also has poor appetite– the body is not free."* (Mother, FGD 1)

Similar to *kakozi*, a few caregivers mentioned how *kapamba* may lead to reduced appetite and the ensuing development of anaemia:

> "Children with kapamba have problems with eating and as they don't eat they get anaemic because blood is produced from different food groups." (Mother, FGD 1)

As with *kakozi*, participants in several FGDs shared the impression that hospital treatment could only temporarily relieve symptoms of *kapamba*. This was especially evident among numerous grandparents who regarded traditional treatment as the only long-term solution for *kapamba*. They described how healers would prepare lumps of medicinal herbs in boiled water before applying it on the affected area. Dark yellow urine was interpreted as a sign of the tumour dissolving or 'breaking' within the body:

> "Traditional healers treat kapamba–they make lumps of medicine and then heat it up on fire and place it on the affected area and some is given orally, and the child urinates yellow stuff which means the tumour breaks inside. We believe in traditional medicine, that is why when a child suffers from something like this we run to traditional healers. We think the hospital does not help, that is why we run for traditional help." (Grandmother, FGD 3)

Both young and old participants acknowledged the existence of *kapamba*, but a larger number of the young participants expressed that they would opt for hospital care. In their view, traditional medicine was used for treating *kapamba* in the past when healthcare services were limited and difficult to access.

## 3.4 Societal disease aetiology

Another reported local illness was *msempho*, which is thought to be caused by adultery. The participants described how *msempho* occurs in a child that has been 'skipped' (*kumusempha mwana*) or 'left' (*kumusiya*) by a parent who is having extra-marital affairs. Subsequently, the child becomes sick and may start imitating sexual intercourse. Thus, the wrongdoings of a parent are believed to be passed on to the child. *Msempho* mostly affects babies, and may be suspected if a baby appears malnourished, weak, pale or has visibly swollen cheeks, hands and feet:

> "I have a grandchild who got sick. We went for traditional medicine and we even went to the hospital. But it did not work until someone told me that by the way the child looks it has been 'skipped'–which means the mother had gone out of the home to do something stupid." (Grandmother, FGD 3)

Both grandmothers and some of the young fathers specified how *msempho* can be transferred to a child through breastfeeding or intake of food that has been prepared–and specifically salted–by the unfaithful parent. However, the majority of participants claimed that adultery-related illness belongs to the past when community elders would, due to 'ignorance', consider certain symptoms as signs of *msempho*:

> "Adultery-related disease in children is an issue of the past. Our parents used to say that a child has been affected by it if it showed signs of swollen cheeks, hands and feet–and they would go for traditional medicine. But these days there are no such diseases, and when a child swells up we say that it is due to a lack of different food groups and we go to the hospital." (Mother, FGD 1)

Several parents did not consider *msempho* as an actual illness, and avidly argued that the tales of adultery-related illness were preached by elders as a deterrent to promiscuous behaviour. In their opinion, these beliefs have resulted in the death of many children that were not given appropriate hospital care:

> *"These are just beliefs in the village–people just speculate and these are things people believed before they knew about hospitals. Many children have died of this in the past because their families have delayed with treatment–many lives have been wasted." (Father, FGD 2)*

Water containing certain herbs and tree leaves given to the child or the unfaithful parent was the only reported treatment option for *msempho*.

### 3.5 Personalistic disease aetiologies

The participants associated anaemia with two categories of supernatural or religious forces, namely witchcraft and Satanism. These theories of ill-health can be considered as personalistic disease aetiologies. Across all discussion groups the participants unanimously agreed that conditions rooted in such forces cannot be adequately treated by the formal healthcare system.

**Witchcraft.** If illness strikes suddenly and occurs repeatedly, community members may suspect the sufferer to be a victim of witchcraft. The caregivers claimed that a child who becomes mysteriously and rapidly ill with no apparent reason or prior warning signs may be struck by the vicious evils of dark magic. If no illness is detected by diagnostic testing at the hospital, or health workers provide a diagnosis but no treatment seems to be effective, some participants were convinced that mystic and malignant forces are at play:

> *"When children are sick they may have bad appetite, and the hospital may test for malaria and not find it, and they say 'we found nothing'. So when we come back home, we start thinking about traditional help. If the hospital did not find anything wrong–it is witchcraft definitely." (Grandfather, FGD 7)*

Recurring anaemia, particularly if anaemia recurs after a blood transfusion, was also linked to witchcraft:

> *"When someone keeps having constant anaemia attacks here at the village we will say that they have been bewitched. If the person has been given blood at the hospital but the blood still keeps going away, then it is the witches sucking out that blood." (Grandmother, FGD 3)*

The participants claimed that illness due to witchcraft has existed in their communities since ancient times. The transfer of illness to a child may happen through food, water, physical touch such as a handshake, or traps (*kasipa*) on roads and farming fields. The evil actions are often driven by 'jealous witches' who envy the wealth, success or material possessions of their victims. Even though the jealousy is mostly triggered by material goods, some participants also shared how healthy and well-fed babies may cause jealousy among witches:

> *"They look at how healthy a baby is, that it is being well-fed and maybe it looks good, so they wish that baby was theirs and then they bewitch it and the baby may die." (Mother, FGD 6)*

Several participants stated that bewitchment of children is often performed to punish their parents:

*"Children are bewitched to make their parents suffer." (Mother, FGD 1)*

According to participants, health workers at the hospital are not capable of diagnosing nor treating witchcraft-related illness:

*"If it is a normal illness you go to the hospital and they give you blood and you will get better, but if it is witchcraft the anaemia attacks will always come back even after being treated." (Mother, FGD 1)*

The participants agreed that only a traditional healer has the necessary tools and training to detect the cause of a witchcraft-related illness and determine the appropriate cure:

*"A traditional healer is the one who knows how to break bewitchment and also for one not to be bewitched again." (Father, FGD 2)*

If formal healthcare does not improve a child's condition, witchcraft is often suspected as the root of the problem–and a traditional healer may prescribe certain herbs for treatment and protection. The herbs can be used in the bathing water, applied to the skin, or be sprinkled in the four corners of the house. There were few or no generational differences among the participants with regards to their perceived association between witchcraft and anaemia.

**Satanism.**    While witchcraft has existed for centuries, a 'new' supernatural or religious force, labelled as Satanism, has reportedly emerged in Malawian communities. It was described by study participants as a modern, 'Western' form of witchcraft that, contrary to 'ordinary' witchcraft, employs tricks of technology and the power of the devil to perform evil deeds:

*"Witchcraft and Satanism are in a way the same, but witchcraft is done out of jealousy, while the satanists will just torment someone or they may just kill someone mysteriously." (Grandfather, FGD 9)*

The participants explained how the 'Western mindset' among satanists is related to their wish for materialistic wealth. Still, witchcraft and Satanism share similarities in the sense that the source of all mischief is in essence supernatural, and the evil acts are performed through magical means. Some participants referred to Satanism as the younger, but stronger, 'sibling' of witchcraft. In their opinion, Satanism holds a markedly greater power than the 'old' witchcraft. The satanists are supposedly members of a group that is working 'against God's will' by causing road accidents and bridge collapses. Members of this group are on a relentless vampire-like quest for blood with the goal of gathering bodily and spiritual strength through human sacrifice. The participants specified how witches are fixed on flesh or so-called 'mortal meat', while followers of Satanism prefer fresh, human blood:

*"After the witches torment someone, they use them as meat. In Satanism, what they want is blood, so they will just take the blood and then leave the body." (Father, FGD 2)*

While witchcraft reportedly takes place during the night, a satanist attack can occur at any time–even in broad daylight. Participation in the satanist network is thought to give great power and wealth, which is a motivating factor for new followers to join. If a person in the community suddenly owns a car or other pricy possessions, he or she may be accused of being a satanist. However, the majority of satanists are unknown since they work in disguise and perform their malevolent acts in secrecy. Young people have an energy level and youthful

appearance that satanists apparently believe will grant them better 'seductive powers'. Hence, the satanists view young people as the most ideal choice of new followers. Also, young people are supposedly more eager to join the network in order to become rich:

*"Our youth are at a higher risk because they want to get rich fast and they do not yet know that they will get the things they want when they get an education." (Grandmother, FGD 4)*

The caregivers shared how satanists drain blood and therefore may cause anaemia in their victims. They listed various methods used by satanists to gather blood, and all of these techniques are invisible to the plain sight of 'normal' people:

*"I heard from my sister who was at boarding school that satanists created bed bugs and these bugs went to their rooms and sucked blood from students. Most of the students were found anaemic and after investigation it was thought to be related to Satanism." (Father, FGD 5)*

One participant expressed how satanists can draw blood from someone simply through staring:

*"It is indeed true that satanists can suck out blood, we have heard that sometimes they may, through their tricks, just suck out the blood by simply staring at the victim." (Grandmother, FGD 3)*

A young mother explained how satanists may enter peoples' houses at night and cause anaemia in children:

*"Sometimes the satanists suck the blood from the child without your knowledge. They use something to penetrate the house and you may live thinking all is well, but in fact the satanists have taken over you, and your child will be sick. At night while you are sleeping they come and suck out blood from the child–when you wake up and go to the hospital the health workers find the child to be anaemic." (Mother, FGD 8)*

It is considered nearly impossible to shield oneself and one's children against the devilish, dark doings of Satanism. Some caregivers highlighted prayer as a strategy for preventing harm and hurt. In their opinion, prayer can serve as a sort of protection against Satanism:

*"The most important thing we do is to pray–there is nothing else that can deal with Satanism." (Father, FGD 2)*

Satanist attacks were also perceived by some participants as a test of a person's faith:

*"The satanists hate prayer and they hate someone who loves to pray. It is also said that their desire is to destroy the belief in God within a person, so if they try and the power of God is too strong then they will let go of you." (Mother, FGD 6)*

During all the FGDs, participants stated that people subjected to Satanism cannot be helped by traditional healers nor health workers at the hospital. In their view, all medical modalities are 'weak' compared to Satanism.

## 4. Discussion

This study found that the local perceptions of childhood anaemia in Zomba, Malawi are multi-faceted. We have used Foster's principles of disease causality in non-Western medical systems [17] and the concept of 'social causes of ill-health' by Kahissay et al. [18] to categorise the different understandings of anaemia causality. The reported 'natural' causes of anaemia include biomedically acknowledged conditions such as malaria and nutritional deficiency. Also included in this group is genital shrinking and bloody urine due to cold weather (*kakozi*), and an organ or tumour-like structure (*kapamba*) which may suck or sequester blood within the body. The local illnesses *kakozi* and *kapamba* are examples of naturalistic disease aetiologies where loss of fluid or heat from the body disturbs its balance or 'humoral equilibrium', and the lack of such a 'bodily balance' is believed to trigger illness [17]. The last of the three local illnesses, *msempho*, has a 'societal' cause. It involves the violation of social norms where infidelity by a parent is thought to cause ill-health in the child. Since this affliction stems from disharmony or disruption in social relations it exemplifies how health and well-being can also be connected to specific actions and behaviours [18, 28]. In Malawi, it has been shown that healing is "commonly oriented to the social body and not solely to the individual body as material object", and that "one task of healers is to diagnose the sites of ruptures in the social milieu and to guide the afflicted in restoring harmony" [29]. Yet, many parents in this study disputed that *msempho* is present nowadays, and stressed that child illness was previously linked to the actions of parents to mitigate unwanted, taboo behaviour such as extramarital affairs.

Finally, the participants talked about witchcraft and Satanism, which we have classified as personalistic disease aetiologies. Within this aetiological domain an 'acting agent' such as a witch, evil spirit or supernatural being is thought to induce illness [17]. In SSA, there are countless reports of how witchcraft is perceived to be the source of ill-health [30, 31], and such accounts are also found in historical records and ethnographic studies conducted in the colonial era. Recent research has found that in rural Senegal "respondents living in neighbourhoods or having social networks with a higher percentage of Christians and those with at least some formal education are less likely to be classified as holding the ethnomedical cultural model" [32]. Still, our study demonstrates how a relatively 'new' personalistic power, Satanism, which is directly linked to the Christian faith, is believed to cause anaemia in children. The idea of satanist groups has, according to participants, appeared more recently and is thus separated from the age-old traditions of witchcraft. Hence, the understanding of illness within the supernatural domain is not fixed or static, and this should be accounted for when we attempt to understand treatment-seeking behaviour.

In this study, there were quite clear generational differences in the perceptions of local illnesses, with parents being more sceptical to some of these illnesses than grandparents. There was however a general agreement among parents and grandparents that the only viable therapeutic alternative for anaemia of supernatural origin is treatment by healers for witchcraft and prayer for Satanism. The coexistence of multiple, competing causal categories is likely to complicate the ability of caregivers to recognise and respond appropriately in cases of childhood anaemia. Especially since findings from this study indicate how each anaemia-related illness is regarded as a separate entity with its own distinct treatment tradition. Thus, the causal reasoning and classification of symptoms will determine what, when and how treatment is sought. This echoes a study among mothers in Gabon where fever of supernatural origin was supposed to be treated by a healer, while fever due to natural causes required biomedical treatment [33].

During episodes of illness in their child or grandchild, several participants described how they would adopt a step-by-step curative pathway where a combination of therapies were tested in a 'multi-try' fashion. This aligns with previous studies where care-seeking for

childhood illness in SSA is described as a dynamic and complex process with various health providers being accessed at different time points depending on the perceived cause and severity of symptoms [4, 34, 35]. It should be noted that several participants would suspect that the child had a local illness or supernaturally caused illness only *after* the formal health system failed to cure the child. In these cases, treatment offered by healers was sought as a secondary strategy if the child's condition did not improve after consulting formal healthcare providers. This reiterates a study from Tanzania where a nurse's son still had anaemia after blood transfusions, and the nurse was advised by some of her colleagues to visit a healer since "the failure of the blood transfusions to 'increase' the child's blood served as a symptom of *mateso*, or witchcraft" [36]. After receiving traditional medicine, she brought her son back to the hospital, but she "did not return to the hospital because the traditional treatment did not work; she returned because the biomedical treatment could only work *after* the sorcerer's mischief was removed" [36]. This example illustrates how supernatural beliefs and the complex terrain of therapies, which are also seen in Malawi, may lead caregivers and sometimes even health workers to make use of both formal and informal treatment.

Malawi is one of the world's poorest countries, and health facilities are seriously under-resourced in terms of staff, equipment and drugs. It is therefore likely that health workers fail to diagnose correctly, or that the child is not given optimal medication. This would not only contribute to morbidity, but also possibly diminish the trust towards formal healthcare and enhance the caregivers' reliance on traditional and complementary medicine. For example, a study performed in a rural village of Malawi found that pregnant women "resisted hospital births, not because they disliked biomedicine, but because the childbirth services were considered to be of such poor quality" and the maternity ward was described as a "place of hostility and lack of care" [37].

Grandparents–and particularly grandmothers–are often actively involved in the childcare decisions at household level in Malawi [24] and other African countries [7, 25, 26, 35]. The views of grandparents may greatly influence the family's treatment choices. In this study, many grandparents preferred to primarily seek help for health issues at community level. They strongly valued the advice of friends, community elders and healers–and would initially use traditional remedies before seeking 'professional' help if symptoms progressed. On the contrary, the parents were especially eager to access hospital care for their children as a primary strategy. This study demonstrates the value of organising FGDs with participants of different age groups to map such generational differences.

In SSA, biomedical treatment is often sought when symptoms worsen or as a 'last resort' [4, 33], but findings from this study reveal how caregivers in Malawi considered traditional and complementary treatment as the only acceptable alternative for some childhood illnesses. Specifically, caregivers emphasised how illness of supernatural origin and certain local illnesses cannot be treated successfully by formal healthcare. This mirrors prior research where healers in Western Africa would treat child illness caused by witchcraft since it 'surpassed the knowledge of doctors' [34].

If the child's illness worsens or reappears despite biomedical treatment, many participants would suspect a supernatural causality such as Satanism. Our study contributes to this field of knowledge by showing that in Zomba, Malawi–and possibly other areas–Satanism should be added to the list of supernatural forces that is perceived to endanger child health.

## 5. Conclusion

The perceived seriousness of the child's condition and its causal classification are important determinants for a family's healthcare-seeking behaviour. In the case of childhood anaemia in

Malawi, caregivers may associate the child's anaemia symptoms with witchcraft, Satanism or a local illness. When caregivers relate the child's anaemia symptoms to these causes it is likely that traditional and complementary treatment at household or community level is sought initially. Findings from this study suggest that multiple, competing theories of causality and treatment, such as in the case of childhood anaemia in Malawi, may lead to critical delays in hospital care of sick children.

## Study limitations

Data collected in this study does not explore actual participant behaviour and practices. The verbal accounts and experiences of participants are self-reported and do not necessarily reflect real-life behavioural patterns. Also, attitude formation and decision-making are inherently unobservable processes which cannot be captured during FGDs [38]. The FGDs included a relatively high number of participants (9–11) with the great majority having 10 participants. The ideal number of FGD participants is debated, and some would argue that smaller groups of 5–8 participants are more ideal [27]. We 'overbooked' slightly due to prior field experience with low turnout, and we did not want to exclude any of those who turned up. However, the dynamics of the FGDs functioned well. While we recruited parents and grandparents of children under five, during the discussions the moderator and participants would use the word "children". We can therefore not differentiate between younger and older children when it comes to the participants' perceptions of anaemia causality. Participants were asked about specific topics guided by previous fieldwork and it is possible that other causes of anaemia would have come up if questions were not directed to these specific topics. There is also a chance that the presence of a European researcher and a middle-class, educated Malawian moderator who was not from the local area resulted in a social desirability effect where participants emphasised their use of public health services. We believe that the way the questions were asked (i.e. mentioning local illnesses without any sign of prejudice and asking them to 'teach' us about them) helped reduce this, and we think it is evident from our results.

## Acknowledgments

We wish to thank Darlen Dzimwe for her valuable work as research assistant and the caregivers for their kind participation. We are also grateful to the health surveillance assistants (HSAs) of the five communities for assisting with recruitment of participants, and to the staff at the Training and Research Unit of Excellence (TRUE) at Zomba Central Hospital for helping us with contact information for the HSAs. Finally, we wish to thank the two anonymous reviewers for their constructive comments, and the project leaders of the main trial which this study is part of, Prof. Kamija Phiri and Prof. Bjarne Robberstad, for their support throughout the research process.

## Author Contributions

**Conceptualization:** Sarah Svege, Siri Lange.

**Data curation:** Sarah Svege, Thandile Nkosi-Gondwe.

**Formal analysis:** Sarah Svege.

**Funding acquisition:** Siri Lange.

**Investigation:** Sarah Svege.

**Methodology:** Sarah Svege, Thandile Nkosi-Gondwe, Siri Lange.

**Supervision:** Thandile Nkosi-Gondwe, Siri Lange.

**Writing – original draft:** Sarah Svege.

**Writing – review & editing:** Thandile Nkosi-Gondwe, Siri Lange.

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
