## [Decision Letter · Decision Letter 0]

27 Jan 2021

PONE-D-20-24587

"Satanism is witchcraft's younger sibling": Changing perceptions of natural and supernatural anaemia causality in Malawian children

PLOS ONE

Dear Dr. Svege,

Thank you for submitting your manuscript to PLOS ONE. After careful consideration, we feel that it has merit but does not fully meet PLOS ONE’s publication criteria as it currently stands. Therefore, we invite you to submit a revised version of the manuscript that addresses the points raised during the review process.

The manuscript has been evaluated by two reviewers, and their comments are available below. You will see the reviewers have congratulated you on addressing the gap in the knowledge around perceived causes of child anemia and related care seeking behavior in Malawi. However, they have also raised several concerns that should be addressed before the manuscript can be further considered for publication.

The key concerns noted by the reviewers relate to the need for more information regarding the inclusion criteria, discussion of the findings in the context of the participant characteristics, and  clarification that the observed associations were a result of asking caregivers about specific topics (and not from open-ended questioning). These limitations have implications for the interpretation of the results and should be explored. 

We look forward to receiving your revised manuscript.

Kind regards,

Danielle Poole

Staff Editor

PLOS ONE

Journal Requirements:

Reviewers' comments:

Reviewer's Responses to Questions

**Comments to the Author**

1. Is the manuscript technically sound, and do the data support the conclusions?

Reviewer #1: Yes

Reviewer #2: Yes

2. Has the statistical analysis been performed appropriately and rigorously? 

Reviewer #1: N/A

Reviewer #2: N/A

3. Have the authors made all data underlying the findings in their manuscript fully available?

Reviewer #1: Yes

Reviewer #2: Yes

4. Is the manuscript presented in an intelligible fashion and written in standard English?

Reviewer #1: Yes

Reviewer #2: Yes

5. Review Comments to the Author

Reviewer #1: A qualitative study (nested in a clinical trial on delivery of malaria chemoprevention as post-discharge management for children with severe anaemia in Malawi) that investigates caregiver (parents and grandparents) perceptions on anaemia causality and treatment options in Zomba, Malawi. All data was collected through FGDs in Zomba district.

A guide of open-ended questions was developed to explore topics of interest. The guide was informed by preliminary knowledge on local perceptions of anaemia from a previous study (2018) by the same study group. Topics of interest were ‘local illnesses that caregivers had associated with anaemia’ (‘kapamba’and‘kakozi’), supernatural forces that caregivers had reported as possible causes of anaemia (‘Satanism’ and ‘witchcraft’), the role of traditional healers’ and the ‘treatment and prevention of anaemia’.

Sickle cell anaemia was an additional topic of interest, but as few participants new of this condition – this was not further discussed.

The results of the FGDs are reported in terms of general perceptions of symptoms and treatment strategies, naturalistic-, societal- and personalistic- disease aetiology – with comparisons between younger parents and elder grandparents.

The authors conclude that caregivers may associate the child’s anaemia symptoms with witchcraft, satanism or a local illness and that if these causes are suspected that it is likely that traditional and complementary treatment at household or community level is sought initially which may lead to delayed hospital care of sick children. .

A well -written paper that addresses gaps in knowledge around perceived causes of child anaemia and related care seeking behaviour in Zomba, Malawi.

Recommendation: minor revision through further clarification of a few unclarities, as further described below.

1. The authors conclude that caregivers may associate the child’s anaemia symptoms with witchcraft, satanism or a local illness (such as kakozi and kapamba). However, participants were asked about these specific topics. It may have been possible that other causes would have come up if the questions would not have been guided to these causes. This may be added as a limitation of the study and/or the conclusion may be better formulated that ‘when’ caregivers associate child’s anaemia symptoms with witchcraft, satanism or a local illness ….. traditional treatment may be sought initially which may lead to delayed hospital care of sick children.

2. Suggest to take out parts re sickle cell anaemia – as it does not add anything of interest.

3. The inclusion criteria for participants are not clear. Why needed the parents to be preferably in the age group 18-35 and was this defined as ‘young’? When reporting about young mothers or young fathers - in which age group were they?

4. Discussion: I would suggest to start with a paragraph that summarizes the results.

5. Conclusion: the ‘hindering of timely hospital care such as iron supplements and blood transfusion’ seems out of place in the Conclusion section – as this is not discussed in the Discussion section.

Reviewer #2: This manuscript explored reasons why rural Malawian children with anaemia might present late to hospital. This was done using focus group discussions with parents and grandparents of children from a rural community in Malawi. The researchers found that the participants distinguished between anaemia due to naturalistic causes, those due to local illnesses and those due to witchcraft or satanism. In general, the participants believed that whilst anaemia due to naturalistic causes could be treated by medical intervention, anaemia due to satanism would only respond to traditional remedies. The study, the first of its kind, gave interesting insight into why children may present late to health facilities, and the complex steps their caregivers take to access different types of care. The manuscript gave a comprehensive summary of the main points that came out of the focus group discussions and were able to put these findings in the context of similar studies from other parts of the world.

This is an interesting paper that I believe will be of interest to the readership of PLoS and should be published, with modifications.

There are a few areas that I believe the authors should address to strengthen the manuscript.

1. Although data on the characteristics and socioeconomic status of the participants were given, these were not discussed further in the context of the findings from the focus group discussions.

a. For example, it would have been interesting to know if there were any differences in beliefs on Satanism in the Moslem participants, given that the paper said that belief in Satanism was directly linked to the Christian faith.

b. Also, were there any differences in health seeking behaviour based on the level of final education of the caregivers?

If the numbers were too small to detect any such differences, it would be worth mentioning, and/or a comment from the authors as to whether religion and/or education (or any other SES factors) may play any role in the decision-making tree of the caregivers.

2. One assumes that the participants were all independent, i.e., they were parents or grandparents of 90 separate children? This should be explicitly stated, and if this was not the case, it should be clear how many participants shared the same child and a statement made about how this might bias the results.

6. PLOS authors have the option to publish the peer review history of their article (what does this mean?). If published, this will include your full peer review and any attached files.

Reviewer #1: No

Reviewer #2: No

---

## [Author Response · Author response to Decision Letter 0]

26 Mar 2021

The response to reviewer and editor comments is given in the attached document titled "response to reviewers".

---

## [Editor Report · Decision Letter 1]

12 Apr 2021

"Satanism is witchcraft's younger sibling": Changing perceptions of natural and supernatural anaemia causality in Malawian children

PONE-D-20-24587R1

Dear Sarah Svege,

We’re pleased to inform you that your manuscript has been judged scientifically suitable for publication and will be formally accepted for publication once it meets all outstanding technical requirements.

Kind regards,

Monique van Lettow, MPH, Ph.D

Guest Editor

PLOS ONE
---

## [Editor Report · Acceptance letter]

23 Apr 2021

PONE-D-20-24587R1 

“Satanism is witchcraft’s younger sibling”: Changing perceptions of natural and supernatural anaemia causality in Malawian children 

Dear Dr. Svege:

I'm pleased to inform you that your manuscript has been deemed suitable for publication in PLOS ONE. Congratulations! Your manuscript is now with our production department. 

Kind regards, 

on behalf of

Dr. Monique van Lettow 

Guest Editor

PLOS ONE